# The Influence of Photoperiod, Intake of Polyunsaturated Fatty Acids, and Food Availability on Seasonal Acclimatization in Red Deer (*Cervus elaphus*)

**DOI:** 10.3390/ani13101600

**Published:** 2023-05-10

**Authors:** Kristina Gasch, Manuela Habe, Julie Sophie Krauss, Johanna Painer-Gigler, Gabrielle Stalder, Walter Arnold

**Affiliations:** Research Institute of Wildlife Ecology, Department of Interdisciplinary Life Science, University of Veterinary Medicine Vienna, 1160 Vienna, Austria; kris.gasch@gmail.com (K.G.); habemanu@gmail.com (M.H.); julie.sophie.92@googlemail.com (J.S.K.); johanna.painer@vetmeduni.ac.at (J.P.-G.); gabrielle.stalder@vetmeduni.ac.at (G.S.)

**Keywords:** body temperature, activity, heart rate, subcutaneous temperature, coat change, circannual rhythms, daylength, melatonin, cervids

## Abstract

**Simple Summary:**

Northern mammals and birds withstand harsh winter conditions by hypometabolism and hypothermia. We investigated in an experimental study with a non-hibernating, but strongly seasonal species, the red deer, whether such seasonal acclimatization is controlled by photoperiod and modulated by the intake of essential polyunsaturated fatty acids (PUFA), like in hibernators. We used advanced data logging techniques to continuously measure heart rate (*f*_H_), a proxy of energy expenditure, core and peripheral body temperature (T_b_), and activity in adult females fed with pellets ad libitum or with restricted daily rations and either enriched with linoleic acid (LA, C18:2 n-6) or α-linolenic acid (ALA, C18:3 n-3). Results show strong seasonal variation of measured parameters, exacerbated by restricted feeding, but with only few, mostly inconsistent, effects of supplementation with LA or ALA. By experimentally increasing the amount of circulating melatonin already in early summer to levels about three times higher than the winter peak, we induced a winter phenotype weeks ahead of time, demonstrating photoperiodic control of physiological and behavioral seasonal acclimatization.

**Abstract:**

Hypometabolism and hypothermia are common reactions of birds and mammals to cope with harsh winter conditions. In small mammals, the occurrence of hibernation and daily torpor is entrained by photoperiod, and the magnitude of hypometabolism and decrease of body temperature (T_b_) is influenced by the dietary supply of essential polyunsaturated fatty acids. We investigated whether similar effects exist in a non-hibernating large mammal, the red deer (*Cervus elaphus*). We fed adult females with pellets enriched with either linoleic acid (LA) or α-linolenic acid (ALA) during alternating periods of ad libitum and restricted feeding in a cross-over experimental design. Further, we scrutinized the role of photoperiod for physiological and behavioral seasonal changes by manipulating the amount of circulating melatonin. The deer were equipped with data loggers recording heart rate, core and peripheral T_b_, and locomotor activity. Further, we regularly weighed the animals and measured their daily intake of food pellets. All physiological and behavioral parameters measured varied seasonally, with amplitudes exacerbated by restricted feeding, but with only few and inconsistent effects of supplementation with LA or ALA. Administering melatonin around the summer solstice caused a change into the winter phenotype weeks ahead of time in all traits measured. We conclude that red deer reduce energy expenditure for thermoregulation upon short daylength, a reaction amplified by food restriction.

## 1. Introduction

At northern latitudes and high altitudes, animals experience profoundly differing living conditions between winter and summer. Temperatures are much lower during winter, forcing endothermic organisms to expend more energy to maintain a high body temperature (T_b_). Simultaneously, especially herbivores are challenged by a nutritional bottleneck, because, outside of the vegetation period, the availability and quality of plant material is low, and food is difficult to access if covered by snow or ice [1,2]. Many small mammals cope with these conditions by hibernation or daily torpor, i.e., by substantially reduced metabolic rate and T_b_ [3]. Interestingly, the ability of these species to tolerate low T_b_ is influenced by the dietary supply of polyunsaturated fatty acids (PUFA) but with apparently opposing effects of n-6 and n-3 PUFA. Diets rich in linoleic acids (LA, C18:2, n-6) promote low T_b_ during torpor, whereas high intake of α-linolenic acid (ALA, C18:3 n-3) has the opposite thermoregulatory effect but is supposed to improve production of adenosine triphosphate (ATP) (reviewed in [4]).

Ungulates of northern latitudes acclimatize to harsh environmental conditions during winter, similar to hibernators or daily heterotherms. They also become hypometabolic [5], as indicated by a major reduction of heart rate (*f*_H_), a good proxy of metabolic rate [6,7,8,9]. This reduction of energy expenditure is to some degree accomplished by reduced locomotor activity, but mainly results from lower endogenous heat production [5]. However, in contrast to torpid or hibernating mammals, core T_b_ is only slightly reduced. In red deer, for instance, the annual variation of rumen temperature (T_r_), which is slightly higher than, but closely follows, core T_b_ [10,11,12], is only in the range of 0.5 °C, but this difference is predominantly responsible for a twofold difference of *f*_H_ at rest between summer and winter [13]. In large ungulates, a slightly lower core T_b_ apparently indicates a much greater reduction of temperature in the body’s periphery [14,15] and, thus, in mean T_b_ of the whole body, the energetically relevant parameter.

Another analogy to the overwintering strategy of small mammals is the shift in ungulates from an anabolic metabolism during summer to the use of body fat reserves during winter as a significant source of energy [5]. This shift leads to a reduction of voluntary food intake (VFI) during winter, persisting even when food is provided ad lib. [13,16,17], hence indicating an endogenous seasonal cycle of appetite entrained by photoperiod [17,18,19,20]. However, limiting the access to food during winter further decreased *f*_H_ and T_r_ in experimentally manipulated red deer beyond the seasonal troughs found with ad lib. feeding [13]. Thus, food availability can amplify or alleviate seasonal changes of metabolism and T_b_.

In the present study, we investigated for the first time in a non-hibernating species, the red deer (*Cervus elaphus*), whether and how the experimental manipulation of the dietary supply of LA and ALA influences seasonal acclimatization on top of changes in food availability. We expected strong seasonality in *f*_H_, T_b_, and locomotor activity, confirming and extending previous results [13] but modulation of the amplitude of seasonal cycles by PUFA supplementation and food availability. Deer fed an LA-enriched diet were expected to tolerate lower T_b_ in winter, both in the body core and shell, resulting in lower *f*_H_ compared to deer fed pellets enriched with ALA, because incorporation of n-6 PUFA into phospholipids (PL) is a well-known phenomenon of cold acclimatization [21] and is the predominant process of remodeling heart PL in Alpine marmots prior to hibernation [22]. On the other hand, deer supplemented with ALA were expected to perform better during the anabolic summer phase because of the accelerated activity of key enzymes of oxidative metabolism in mitochondria with PL containing high concentrations of n-3 PUFA derived from ALA [23,24,25]. Due to the hypothesized effects of dietary PUFA on an animal’s ability to cope with energetic challenges, we expected, during periods of food restriction, even higher differences between LA- and ALA-supplemented deer in the aforementioned reactions.

Lastly, we wanted to test whether all aspects of seasonal acclimatization measured are governed by photoperiod, i.e., respond to experimental administration of melatonin during the summer by a phase-advance of the annual rhythm. While this is well-known for the seasonal cycles of VFI, body weight, molt, antler growth, and the onset of the breeding season (reviewed in [26]), endogenous control of seasonal cycles of metabolic rate, T_b_, and locomotor activity is so far not experimentally confirmed for red deer.

## 2. Materials and Methods

### 2.1. Experimental Animals and Feeding

We used for this study adult female red deer (*Cervus elaphus*) living in a 45-hectare (ha, 45,000 m^2^) enclosure adjacent to the Research Institute for Wildlife Ecology, Vienna (48.21° N, 16.37° E, 360 m above sea level). The enclosure consisted of ~39 ha mixed beech and oak forest and a ~6 ha meadow. The experimental animals belonged to a bigger herd containing 21 adult females, progeny from several years, and an adult stag. The animals were individually marked with ear-tags with transponders and individually colored collars.

The study was conducted from March 2017 until March 2020. Prior to the study, we selected 16 adult females, age 2–9 years, and randomly assigned them to two feeding groups of eight individuals each, balanced by age and body mass and, thus, also by social rank [27]. These females produced during the study 42 calves, a rate of fertility of 96%. The experimental females received food pellets at an automated feeding station (Schauer, Prambachkirchen, Austria), in addition to the natural forage available in the enclosure. Anteceding the study, the deer were familiarized with this station. The station delivered one of two pre-programmed types and daily amounts of pellets and registered individual daily consumptions.

The pellets supplied were commercially available red deer pellets (Garant Trophy STTM Luzerne Apple, Raiffeisenverband Salzburg, Austria) immersed in 10% *w*/*w* linseed oil or sunflower seed oil, to supplement predominantly with either ALA or LA, respectively. After treatment, both types of pellets remained nearly iso-caloric (Table 1). The high fat content of the pellets led to signs of obesity in many animals after one. We therefore reduced the amount of oil added to 5% *w*/*w* for the rest of the experiment (Table 1).

To ensure that each animal could be used as its own control, we swapped the type of oil supplementation between experimental groups every year in June. Further, we regularly alternated monthly periods of ad lib. and restricted feeding. During restricted feeding, each individual received per day only 20% of the daily number of pellets consumed during the preceding ad lib. period. For more technical details concerning the experimental feeding and calculations of daily LA and ALA intake, see [28].

### 2.2. Biologging

All experimental female red deer were equipped with telemetry collars (Vectronic Aerospace) with a self-constructed add-on for measuring *f*_H_ and T_r_. One of the collars batteries was replaced with an electronic device storing data, measuring activity, and communicating with a cylindrical probe (22 × 80 mm, 100 g) introduced into the rumen with an applicator during reversal of an anesthesia (see Section 2.4) when the swallowing reflex had resumed. This probe typically settled in the reticulum, in close proximity to the heart, and continuously measured *f*_H_ with a bi-axial acceleration sensor and T_r_ via a thermistor with a precision of 0.1 °C. The thermistor was calibrated in a water bath before the administration of the probe, at 5 °C increments between 20 and 40 °C. T_r_ was measured every three minutes, and *f*_H_ was determined with an adaptive trigger level during periods of three minutes. *f*_H_ was calculated from the time interval between two subsequent activations of the acceleration sensor above the trigger level. Because the measurement of *f*_H_ was very battery draining, the following measurement period started after a pause of four minutes. The ruminal unit transmitted *f*_H_ and T_r_ data via an UHF link to the collar, where it was stored in a solid-state memory. Further, we recorded subcutaneous temperature (T_sub_) every five minutes with implanted temperature loggers (22 × 36 × 7 mm, 8 g). Under surgical anesthesia, an approximately 2 cm incision was made craniolateral of the Manubrium sterni to place the logger subcutaneously. The subcutis and skin were closed using the synthetic absorbable surgical suture material USP 0 (Surgicryl PGA, SMI AG, Hünningen, Belgium). Loggers were calibrated in a water bath before implantation at 5 °C increments between 5 and 35 °C, with an accuracy of 0.1 °C, and data were stored on-board in a solid-state memory. Loggers were replaced halfway through the experiment, when the storage capacity was reached.

Movements of an animal were detected with a tri-axial accelerometer located in the collar. The distribution of the sum of tri-axial outputs during a three-minute period was bimodal, with peaks close to 0% and 100% activation. The lowest point of the trough between peaks was set as the cut-off point, to define an animal’s behavior during a three-minute interval as “active” or “inactive”.

To assess ambient temperatures relevant for the animals, we measured the temperature every three minutes with a thermistor in the black instrument and battery case of the collar under the animals’ heads (T_c_, accuracy 0.5 °C). These measurements were virtually identical and correlated closely with air temperatures measured every ten minutes at a weather station located in the enclosure at a height of 4 m (R^2^ = 0.96). Only in the sub-zero range, T_c_ tended to be slightly higher than air temperature. Activity and T_c_ values were also stored in the collar’s solid-state memory.

### 2.3. Administration of Melatonin

From 27 May 2019–2 July 2019, we subcutaneously implanted in each experimental animal a melatonin depot, next to the subcutaneous temperature logger or on the opposite site of the neck. The depots were silicone membrane bags (Dow Corning Silicone elastomer membrane, 7-4107, size 4 × 4.5 cm), sealed with a silicone adhesive (Dow Corning, Silastic Medical Adhesive Silicone, Type A), and contained 1 g of melatonin (Alfa Aesar, AAJ6245206, Haverhill, MA, USA). By constant diffusion of melatonin through the silicone membrane, we increased the level of circulating melatonin, beginning with early summer, to an estimated concentration of 354 pg.mL^−1^ blood over a period of at least three months (for details on underlying calculations, see [28]). This concentration was about three times higher than the peak winter values reported for Iberian red deer (*Cervus elaphus* hispanicus, [29]).

### 2.4. Anaesthesia

All surgeries and introductions of rumen loggers, along with muscle biopsies, were carried out under anesthesia and post-surgical preemptive analgesia [28]. For immobilization, the deer were called and guided into a wooden corral (wall height 2 m) located in the enclosure. We familiarized the animals with this procedure prior to the study to ensure stress-free handling and anesthesia induction. The anesthesia was induced with medetomidine (0.1 mg.kg^−1^; Medetomidine-HCL 2%, magistral formulation, Richter Pharma AG, Vienna, Austria) combined with tiletamine-zolazepam (3 mg.kg^−1^; Zoletil100, Virbac, Österreich GmbH, Wien, Austria) administered intramuscularly (IM) via hand or blow pipe injection (BLOW 1.25 Model Zoo; Dan-Inject, Kolding, Denmark). During the entire anesthesia, the animals were administered 100% oxygen (2l min^−1^), and their vital parameters (e.g., *f*_H_, mean arterial blood pressure, respiration rate, end-tidal CO_2_, arterial oxygen saturation, rectal T_b_) and sufficient depth of the surgical anesthesia were monitored. The anesthesia was terminated approximately 60 min after induction by IM administration of antagonizing medetomidine with atipamezole (5 mg for each mg of medetomidine; Narcostop, Richter Pharma AG, Wels, Austria). Each animal was monitored continuously until fully recovered and checked for signs of re-narcotization several times during the following 24 h [30].

### 2.5. Measuring Coat Change

We evaluated the time course of coat change from pictures taken weekly from every individual during the periods of change from the summer into the winter coat in 2018 (control) and 2019 (melatonin treatment). The individuals were identified on the pictures by individual characteristics and markings. To ensure standardized assessment, the pictures were analyzed in random order, in the same room, under the same source of light, and with the same computer screen.

We only analyzed images when at least four body compartments could be safely evaluated (Figure 1). The state of shedding was characterized for each compartment on an ordinal scale from 0–6 (0 = still complete summer coat; 1 = lose hair in some parts but less than in a third of the compartment; 2 = a single area of completely changed fur visible; 3 = lose hair in more than a third of the compartment; 4 = more than one area of completely changed fur visible but less than a third of the whole compartment; 5 = completely changed fur in more than a third of the compartment; 6 = completely changed fur in the whole compartment).

### 2.6. Data Analyses

Prior to statistical analyses, we discarded logged data considered unreliable. These were (i) all data obtained during one day before and after an anesthesia or other medical treatment and data obtained during periods close to battery death; (ii) *f*_H_ measurements, if the signal from the acceleration sensor did not exceed the trigger level during less than 70% of a three-minute measurement interval; (iii) *f*_H_-measurements when an animal’s behavior was classified as “active”, because animal movements also activated the acceleration sensor in the ruminal unit. For each of the remaining three-minute intervals of *f*_H_ measurements, we computed a Kernel density function for the *f*_H_ values recorded and used the *f*_H_ at maximum density as a value for a given interval. Finally, we fitted a spline curve to all *f*_H_ values determined over the period of deployment of a telemetry system and discarded values exceeding the spline fit prediction by more than 20 beats.min^−1^. Because the remaining data represent *f*_H_ measured at rest or low activity, we refer to it in the following as stationary heart rate (*f*_Hs_, [13,31,32]). Temperature data contained exceptional low but rapidly changing values, probably due to the ingestion of cold water or food in the case of T_r_, or to wallowing in the case of T_sub_ and T_c_. To remove these data, we discarded all values differing from the previous value by more than 0.1 °C (T_r_), 0.5 °C (T_sub_), or 1 °C (T_c_), respectively.

For statistical analyses, we used R version 4.2.2 [33]. Over the period from the beginning of the study until the beginning of the melatonin experiment we modeled intake of pellets, of LA and ALA, locomotor activity, T_sub_, T_r_, *f*_Hs_, and T_c_ with generalized additive mixed-effects models, using the function “bam” for large datasets. We fitted these variables as responses and time as thin-plate regression splines, with the smoothing parameter k = 8 [34], separately for four different feeding regimes (two types of oil supplementation, pellet availability ad lib. or restricted, see above), with the same smoothing parameters, using the smoother “fs” to model interactions. To account for unknown differences between years, and for repeated measurements of individuals, we included experimental year (first year June 2017–May 2018, second year June 2018–May 2019) and experimental year nested within individual ID as random effect smooth terms. Further, for variables measured with high temporal resolution (activity, T_sub_, T_r_, *f*_Hs_, T_c_), we included time of the day as a cyclic P-spline to account for rhythmic daily variations. Lastly, in all models except T_c_, we accounted for temporal autocorrelations among repeated measurements of individuals within periods of unchanged feeding regime by applying an autoregressive function. We used the final models for calculating 95% confidence interval belts for each fit.

Data from melatonin-treated animals were available from 27 May 2019 to 8 February 2020. Except for coat change data, we selected as control data from the same individuals gathered during the same period of the year but two years earlier, when each experimental animal had received the same oil supplementation. These two groups were compared similarly, as described above, for the four feeding regimes but with smoothing parameter k = 6 and only individual ID as a random effect smooth term. For pellet intake as a response variable, we used only data from periods of ad lib. feeding. For analyzing coat change data, we used the R-package “ordinal” [35] with a score of coat change in a body compartment as the response, and week of the year, control vs. melatonin treatment, and type of oil supplementation as predictors in the model. To account for repeated measurements, we modeled body compartment nested within individual ID as the random effect.

## 3. Results

### 3.1. Seasonal Changes of Behavioural and Physiological Parameters

We choose the time span shown in Figure 2 for comparing the four feeding regimes to guarantee identical treatments for intra-individual comparisons, i.e., beginning with the first swap of oil supplementation in June 2017 and ending just before implanting the melatonin depots.

The daily intake of pellets under ad lib. feeding was, during 2017, much higher than during subsequent years and particularly low during autumn 2018, presumably due to the mass fructification of beech and oak in the enclosure in 2018. As a result, the daily intake of LA and ALA with pellets by the experimental animals varied accordingly over time, but with profound differences caused by the type of oil supplementation (sunflower seed oil vs. linseed oil, Figure 2).

All other parameters measured showed less-pronounced differences between the four feeding regimes but stronger cyclic seasonal variation (Figure 2). Locomotor activity always peaked in June. Winter troughs occurred at the end of January, but already in December in food-restricted animals during winter 2018/2019. When pellets were available only at restricted rations, locomotor activity was significantly higher throughout the time span shown in Figure 2, except for January/February during winter 2017/2018 and November to January during winter 2018/2019 (Figure 2).

The seasonal course of T_r_ was like that of activity but with virtually no consistent differences between the four feeding regimes and less pronounced troughs during winter 2018/2019, as in T_sub_ (Figure 2). The troughs of T_sub_ were significantly lower in LA-supplemented animals in winter 2017/2018. Further, during winter 2017/2018, food-restricted animals had lower T_sub_ than animals fed ad lib. for both types of oil supplementations (Figure 2). During winter 2018/2019, a different and less clear pattern with, again, no consistent effect of the type of oil supplementation was present also in T_sub_. Animals fed ad lib. reached the winter trough in 2018/2019 earlier. In animals receiving only restricted daily rations, the linseed oil-supplemented group reached the winter trough in February, like in the year before, but no clear winter trough could be identified in the sunflower seed oil-supplemented group (Figure 2).

The seasonal courses of *f*_Hs_ largely reflected the energy-consuming processes of T_b_ regulation and, with respect to seasonal changes, also that of locomotor activity. However, food-restricted animals had consistently lower *f*_Hs_ despite higher locomotor activity. Surprisingly, during winter 2017/2018, *f*_Hs_ in LA-supplemented animals was slightly higher, like T_r_, despite opposite trends in T_sub_. Again, no significant differences between types of oil supplementation were found during winter 2018/2019, but troughs were again less pronounced compared to winter 2017/2018 (Figure 2).

### 3.2. Photoperiodic Control of Seasonal Changes

To corroborate photoperiodic control of the seasonal changes of VFI, locomotor activity, T_sub_, T_r_ and *f*_Hs_, we disabled the physiological perception of daylength by administering a constant and continuous influx of melatonin. After implantation in early summer of the last study year, the diffusion of melatonin out of the subcutaneous depots in the experimental animals, despite long summer days, led to estimated levels of circulating melatonin three times higher than the peak winter values reported for Iberian red deer (*Cervus elaphus* hispanicus [29], see Section 2.2).

To analyze the effects of melatonin treatment, we compared the seasonal cycles of the measured parameters with their course during the same period two years before. For VFI, we restricted our analysis to periods of ad lib. feeding. As intended, the administration of melatonin caused a phase advance of the annual cycle in all the animal parameters measured, although the course of T_c_, reflecting ambient temperature, was virtually identical during both years. If anything, during autumn/early winter, T_c_ was even higher in the melatonin treatment year (Figure 3). Immediately after implantation of the melatonin depots, VFI dropped to a level typical for winter/spring in the control year. However, VFI increased again towards a late summer peak, like in the control year, but peak consumption remained about 3 kg.day^−1^—significantly below the value of about 4.5 kg.day^−1^ found in the control year (Figure 3). Locomotor activity also declined to the low winter level faster in melatonin-treated animals than in the same animals during the control year. Similar patterns were evident in the seasonal courses of T_sub_ and T_r_. As a result of these energy-consuming processes, *f*_Hs_ also declined faster and reached low winter values about two months earlier than in the control year. Interestingly, reversing the declines after the winter troughs took place earlier in melatonin-treated animals in all parameters measured, except VFI (Figure 3).

### 3.3. Factors Influencing Coat Change

To corroborate that our melatonin administration indeed blurred the photoperiodic measurement of daylength in the expected manner, we also measured the course of change from the summer into the winter coat. It is well known that this change is phase-advanced in animals treated with melatonin [17,18,20]. During the period of coat change, the applied shedding score increased, on average, by 0.81 per week (Figure 4; standard error of the estimate [s.e.], 0.03, z-value = 24.9, *p* < 0.001). However, during 2019, when the animals had subcutaneous melatonin implants, the change into the winter coat occurred, on average, 4.2 weeks earlier compared to 2018 (s.e. 0.23, z = 17.8, *p* < 0.001). There was also a significant effect of the type of oil supplementation, similar in both years. Animals supplemented with ALA changed faster into the winter coat (−0.65 weeks, s.e. 0.13) compared to animals receiving LA supplementation in the same year (Figure 4; z = −5.0, *p* < 0.001).

## 4. Discussion

### 4.1. Effects of Food Supplementation with Polyunsaturated Fatty Acids

Our results confirm previously reported strong seasonality in the measured physiological and behavioral parameters in red deer [13]. However, with respect to the major question of this study—whether the dietary supply of LA promotes the winter depression of *f*_H_ and T_b_, whereas supplementation with ALA has opposing effects—we found no clear answer. During the first experimental year, some of the hypothesized effects of LA supplementation with sunflower seed oil and of ALA supplementation with linseed oil were found. As hypothesized, during winter 2017/2018, T_sub_ was lower in animals supplemented with LA. Surprisingly, the opposite trend was present in T_r_ and *f*_Hs_. Thus, in this experiment, *f*_Hs_ seemed to be more influenced by T_r_ than by T_sub_. However, all of these trends were absent during winter 2018/2019. There are, in our view, four possible, not mutually exclusive, explanations: (i) The dietary supply of LA and ALA has limited and no immediate physiological consequences. This is, in our view, the most likely reason, because there is no correlation between the concentration of LA, or the LA-derived arachidonic acid in phospholipids, and the mean daily LA intake during the two weeks prior to sampling of muscle tissue for the analysis of fatty acids concentrations in PL. Conversely, such a relation was found for ALA, but only when ALA intake was high and, again, for none of the ALA-derived n-3 PUFA [28]. (ii) Our experimental PUFA supplementation was unbalanced, despite adding equal amounts of sunflower seed and linseed oil, because the base pellet contained about four times more LA than ALA. (iii) Oil immersion of pellets caused an overload with the dietary supply of LA and ALA, as indicated by a trend towards obesity in the experimental animals. (iv) The pronounced mast of beech and oak in the study enclosure during the second study year likely spoiled the intended different supply of LA. Seeds such as beech and oak nuts are rich in LA [36]. The excessive availability of these fruits, a preferred feed of red deer during autumn and winter, led to a considerably lower daily intake of food pellets in 2018 compared to 2017 and presumably levelled the supply of LA to the two experimental groups.

However, we did find the expected effect of ALA supplementation on coat change. Molt is an energetically costly process that is delayed or takes longer in energetically constrained animals [37,38,39,40,41]. The change from the summer into the winter fur was completed in a shorter time in animals supplemented with ALA compared to those supplemented with LA. This was the case in both the control year and the year with earlier change of coat due to administering melatonin around the summer solstice. The likely explanation is that supplementation with ALA translates into a corresponding increase of ALA in mitochondrial PL, and this accelerates the activity of membrane-bound key enzymes of ATP production [28]. Thus, this enabled faster growth of the new coat.

Reactions to food restriction also confirmed the already-reported modulating effect of food availability on seasonal acclimatization (cf., [13]). When access to food pellets was restricted, the deer reacted with lower *f*_Hs_ and T_b_ compared to ad lib. Feeding, the latter most pronounced in the body’s periphery. In contrast, during the summer months, locomotor activity was even higher when pellet supply was restricted. During the period of fattening, the deer apparently compensated for the severe reduction of pellet availability by increased foraging on natural vegetation. This was only possible with a considerable increase of movements in the large enclosure in search of food. Interestingly, this increase of locomotor activity did not increase *f*_Hs_. Energy expenditure, as reflected by the proxy *f*_Hs_, remained significantly lower during periods of food restriction, indicating that savings due to lower T_b_, particularly in the body’s periphery, more than compensated for the energetic cost associated with acquiring natural forage.

### 4.2. Effects of Treatment with Melatonin

A winter depression of metabolic rate at the cost of lower T_b_ is apparently a ubiquitous reaction in mammals adapted to habitats with cold winters. This is well known for hibernators and daily heterotherms [42]. However, it has also been found in a number of non-hibernating species, such as ungulates [5,14,15,31,32,43,44,45,46,47,48,49], (but see [50]) as well as in many other mammals (e.g., [51,52,53,54,55]) and birds (e.g., [56,57,58,59]). Less clear is whether the winter decrease of energy expenditure and T_b_ in deer is endogenously controlled like cycles of molt, antler growth, reproduction, and VFI [26], or whether it simply results from lower VFI and, hence, energy expenditure necessary to search and process food and the associated lower heat increment of feeding [60,61,62]. Although more recent work rebutted this view [13,14,32], an ultimate experimental verification is so far lacking. Results from our melatonin experiment now show that the premature decline of *f*_Hs_ in melatonin-treated animals was mainly due to a lower energetic cost for maintaining high T_b_. T_r_ decreased simultaneously with *f*_Hs_, and both decreases occurred despite the concurrent increase of pellet intake (Figure 3). The earlier reversion into the summer phenotype of melatonin-treated animals after about seven months corresponds to the well-known phenomenon of photorefractoriness. Under constant photoperiod, as pretended by constant and high levels of circulating melatonin, the interaction between a circadian-based, melatonin-dependent timer that drives the initial photoperiodic response and a non-circadian-based timer that drives circannual rhythmicity dissociates and leads to the apparent refractory state [63].

## 5. Conclusions

Seasonal cycles of physiological and behavioral traits, with troughs during winter, are known to exist in numerous species inhabiting zones with harsh winter conditions. It appears that these cycles are ubiquitously governed by an ancient endogenous circannual rhythm entrained by photoperiod. This mechanism is responsible for preparing, in a timely manner, not only hibernators and daily heterotherms but also many non-hibernating species, for the profound change of living conditions in their seasonal environments. With our study, we corroborate for the first time— experimentally, for a non-hibernating large mammal, the red deer—that seasonal changes of T_b_ and the resulting energy expenditure, measured with the proxy *f*_Hs_, are also governed by the same mechanism. Conversely, with our feeding experiment, we were not able to answer unequivocally and comprehensively whether and how the dietary supply of essential fatty acids influences seasonal acclimatization at the organismic level.

## Figures and Tables

**Figure 1 animals-13-01600-f001:**
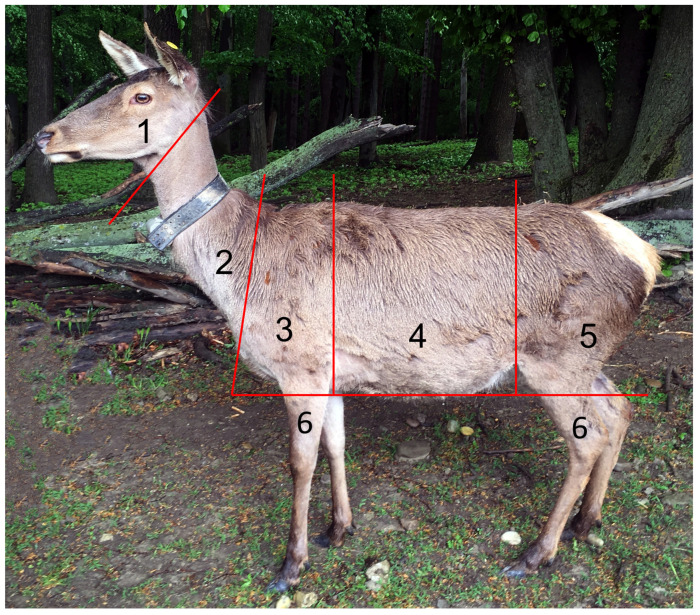
Apportionments of the deer body defined for assessment of coat change. The state of coat change was assessed at six different body compartments: **1** head—from the tip of the nose to the vertical line between the base of the ear and ventral edge of the mandible; **2** neck—from the vertical line between the base of the ear and ventral edge of the mandible to the line between the cranial edge of the withers and cranial line of the breast muscles; **3** thorax—from the line between withers and the cranial breast muscles to the caudal edge of the shoulder blade and elbow, excluding extremities; **4** abdomen and back—from the caudal edge of the shoulder blade to the vertical line of the knee fold; **5** croup and flank—from the vertical line of the knee fold to the tip of the tale, excluding extremities; **6** extremities—from the horizontal line of the elbow/knee to the claw.

**Figure 2 animals-13-01600-f002:**
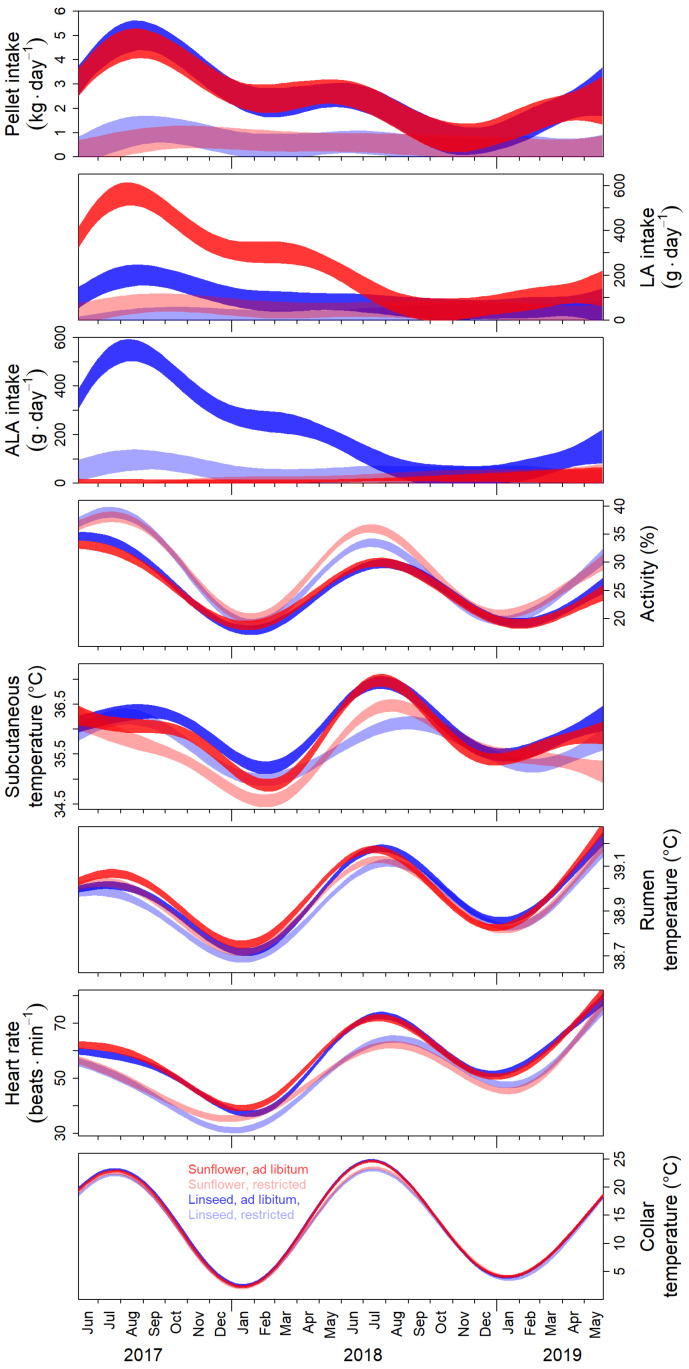
Seasonal changes of behavioral and physiological parameters and effects of type of oil supplementation and pellet availability. The period shown is from the first swap of oil supplementation in June 2017 until the beginning of the melatonin treatment at the end of May 2019. During this period, every experimental animal was exposed to each of the four feeding regimes: sunflower seed oil (red) vs. linseed oil supplementation (blue), ad lib. (low-transparent) vs. restricted feeding (high-transparent). We show 95% confidence belts for the four feeding regimes of predictions from fits across all individuals with a generalized additive mixed-effects model, as functions of time with the individual and experimental year (1st vs. 2nd) as random effect smooth terms. Non-overlapping confidence belts indicate significant differences.

**Figure 3 animals-13-01600-f003:**
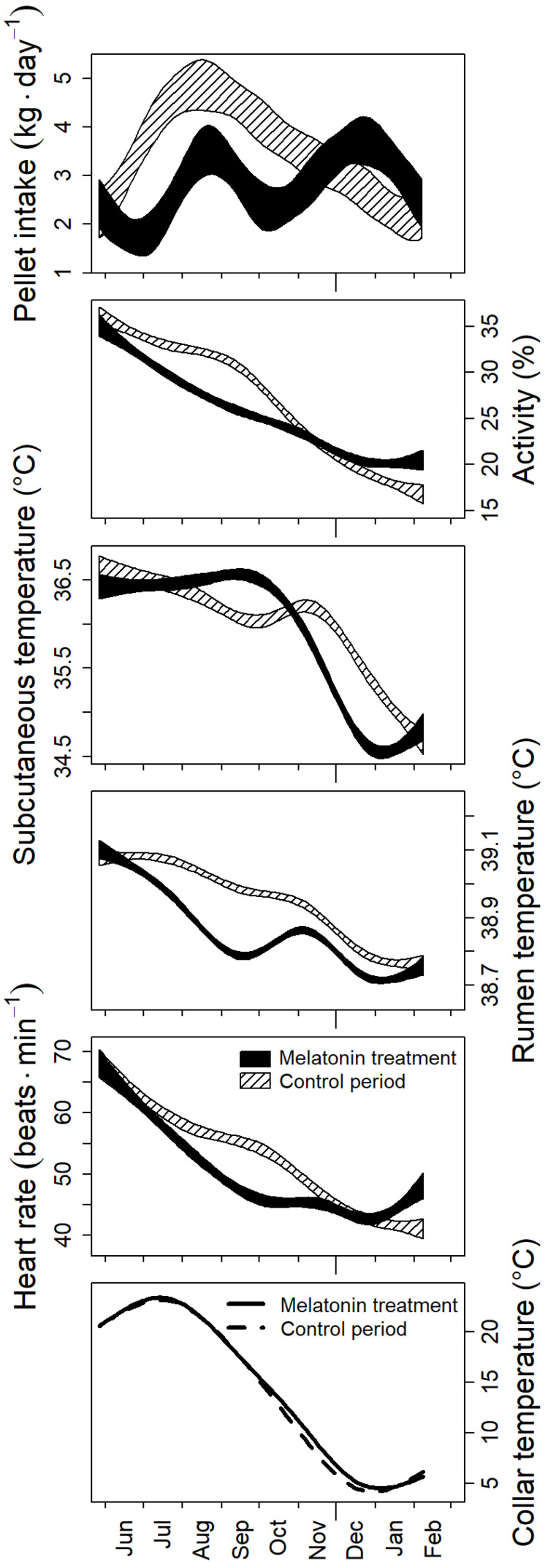
Effect of melatonin treatment on seasonal changes of behavioral and physiological parameters. Periods shown are from the beginning of melatonin administration in June 2019 until the end of data acquisition at the beginning of February 2020 (melatonin treatment) and the same period of the year two years before, when the experimental animals had received the same type of oil supplementation (control period). For pellet intake, only data from periods of ad lib. feeding is shown. Plotted are 95% confidence belts of predictions for the melatonin-treatment (black) and control periods (hatched) from smooth fits across all individuals with a generalized additive mixed-effects model, as functions of time, with the individual and experimental year as random effect smooth term. Data of collar temperatures are means of smooths, because 95% CI were too narrow for visual discrimination (control period: solid line, melatonin treatment period: broken line).

**Figure 4 animals-13-01600-f004:**
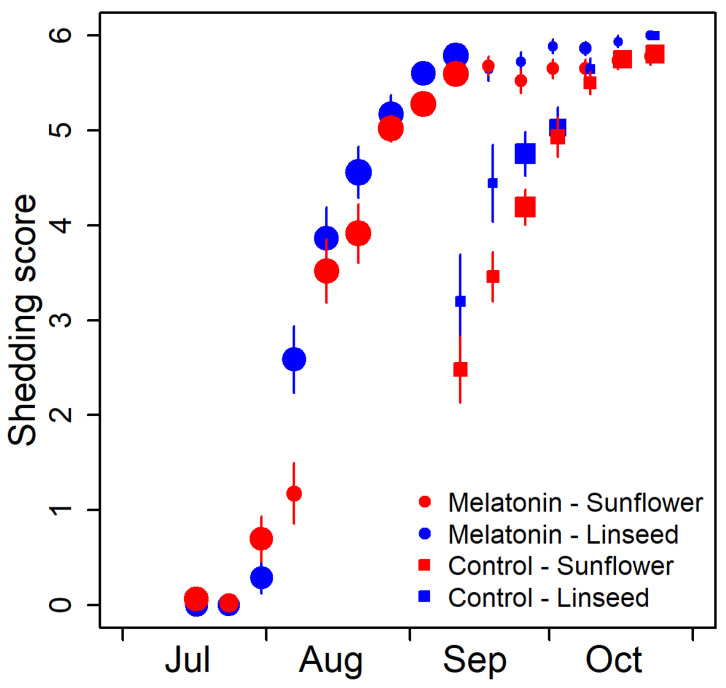
Effect of melatonin treatment on the time course of the change from summer into winter coat. The mean shedding scores (± standard errors of the mean) are plotted as circles for animals with subcutaneous melatonin implants in 2019 and as squares for the coat change of the same animals evaluated during the year before. Colors indicate the type of oil supplementation (red: sunflower seed oil; blue: linseed oil). The size of the symbols reflects the sample size.

**Table 1 animals-13-01600-t001:** Composition and nutritive value of pellets supplied.

Type of Pellet	Crude Fat (%)	Crude Protein (%)	NDF (%) +	ADF (%) *
Immersed in 10% *w*/*w* linseed oil	23.6	12.8	30.9	19.6
Immersed in 10% *w*/*w* sunflower seed oil	21.8	11.9	30.8	19.2
Immersed in 5% *w*/*w* linseed oil	7.9	12.9	31.0	20.6
Immersed in 5% *w*/*w* sunflower seed oil	7.8	16.3	31.4	19.4

+ neutral detergent fiber; * acid detergent fiber.

## Data Availability

Publicly available datasets were analyzed in this study and are available on Phaidra (https://doi.org/10.34876/myfe-ab40, accessed on 5 May 2023).

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
