# Peer review of "The Influence of Photoperiod, Intake of Polyunsaturated Fatty Acids, and Food Availability on Seasonal Acclimatization in Red Deer (Cervus elaphus)"

_animals, 2023, doi:10.3390/ani13101600_

Round 1

Reviewer 1 Report

The main comments on this article are as follows:

1. I recommend changing the title to "Intake of Polyunsaturated Fatty Acids, and Photoperiod Influence Seasonal Acclimatization of Red Deer (Cervus elaphus)" 

2.Did you draw blood during the experiment? Seasonal acclimatization affect the physiology of mammals more in terms of physiological and biochemical indicators of blood than changes in activity, body temperature, heart rate, etc. There are many studies to refer to, such as black and brown bears. It is possible that differences in the intake of polyunsaturated fatty acids can lead to changes in some enzymes in the blood.

3.If possible, please put the photo that how assessed the red deer at six different body compartments into the article, it is more visual and concise than text description.

4.There is no figure 2. Please add.

5. The icon should be below the figure, please adjust it.

6. For aesthetics and consistency, please adjust the font of Figure 3 to be the same as Figure 1.

7.The discussion needs to be strengthened and made more structured. The first paragraph containing the influence of intake of polyunsaturated fatty acid on physiological and coat change in red deer. It is suggested to discuss in segments.

8.The first and two paragraphs suggest citing more references or essay to verify the conjecture.

9.The section on the effects of polyunsaturated fatty acids and melatonin on the coat suggests additional discussion.

10.The NO.26 essay has no doi. Please add.

Author Response

  1. I recommend changing the title to "Intake of Polyunsaturated Fatty Acids, and Photoperiod Influence Seasonal Acclimatization of Red Deer (Cervus elaphus)"

We decided to follow this recommendation only partly because the suggested title does not entirely capture the take-home message of our paper. Our emphasis is on different importance of photoperiod, intake of polyunsaturated fatty acids, and food availability on seasonal acclimatization in red deer. However, to better express our view about the order of importance of these factors, we rearranged the phrasing accordingly.

2.Did you draw blood during the experiment? Seasonal acclimatization affect the physiology of mammals more in terms of physiological and biochemical indicators of blood than changes in activity, body temperature, heart rate, etc. There are many studies to refer to, such as black and brown bears. It is possible that differences in the intake of polyunsaturated fatty acids can lead to changes in some enzymes in the blood.

The reviewer refers here to an important point. In this study, we did not sample blood but muscle tissue. Indeed, there are major seasonal changes in the fatty acid composition of phospholipids which again influence the activity of membrane bound enzymes. However, we decided to present these important aspects of seasonal acclimatization on the molecular level in a separate paper, to be submitted soon to the MDPI journal “Biology”. Here we would like to restrict our analyses to effects on the organismic level, i.e. on activity, body temperature, heart rate, and change of coat.

3.If possible, please put the photo that how assessed the red deer at six different body compartments into the article, it is more visual and concise than text description.

Done

4.There is no figure 2. Please add.

Sorry, the Figure must have been lost during the submission process. The figure was included in the file uploaded. We will now upload the figures not only in the text but also as additional files and ask the publisher to make sure that all figures are correctly included.

  1. The icon should be below the figure, please adjust it.

We agree with this objection. However, we tried to adjust the text properly in the WORD template provided by “animals”, but the position of Figures apparently changes when uploading the file. We ask the publisher to correctly place Figure titles and legends in the text during final editing. We will double check correct alignment when receiving page proofs.

  1. For aesthetics and consistency, please adjust the font of Figure 3 to be the same as Figure 1.

We agree with this objection. In fact, we used the same font for all Figures. Figure 3 needs to be shrunk for publication. We ask the publisher to shrink or expand during final editing all Figures of the paper in a way to achieve identical size of the fonts. We will double check whether this occurred satisfactorily when receiving page proofs.

7.The discussion needs to be strengthened and made more structured. The first paragraph containing the influence of intake of polyunsaturated fatty acid on physiological and coat change in red deer. It is suggested to discuss in segments.

We structured now the Discussion in with two subheadings 4.1. Effects of food supplementation with polyunsaturated fatty acids, and 4.2. Effects of treatment with melatonin. We further structured 4.1 with separating the Discussion of effects of PUFA on coat change into an own paragraph.

8.The first and two paragraphs suggest citing more references or essay to verify the conjecture.

We added a sentence with citations underlining that the change of coat is an energetically costly process. Therefore, accelerated change of coat likely resulted from increased ATP production in mitochondria containing more ALA in phospholipids. We show the latter in a separate paper submitted to MDPI “Biology”. This paper is cited.

9.The section on the effects of polyunsaturated fatty acids and melatonin on the coat suggests additional discussion.

See answer to 8.

10.The NO.26 essay has no doi. Please add.

No. 26 was indeed a wrong citation. We replaced it with the correct citation “Lincoln and Richardson, 1998” and provide also a doi.

Reviewer 2 Report

GENERAL COMMENT:

I consider this work is within the scope of “Animals”. It contains information useful in a field in which available information is scarce. Overall, it is well written and organised. I indicate below only minor points to be improved in the manuscript.

TITLE:

It is all right.

SIMPLE SUMMARY:

It is OK.

ABSTRACT:

Remove “(1)”, “(2)”, “(3)” and “(4)” in lines 22, 26, 31 and 35.

The Abstract must be understood independently of the other parts of the article, because it is also included in the indexing databases. Therefore, it is necessary to write with the full words the first time the following symbols and acronyms are written:  Tb, PUFA, LA, ALA, fH

KEYWORDS:

I recommend adding the following keyword: “cervids”.

INTRODUCTION:

Overall, this section is OK.

Line 52: The first time “ATP” appears in the text, must be accompanied by the full words, with ATP between brackets: “adenosine triphosphate (ATP)”.

Line 83: The first time “PL” appears in the text, must be accompanied by the full words, with PL between brackets.

MATERIALS AND METHODS:

Overall, this section is OK.

Line 114: Add the information of the proximal composition and nutritive value of the red deer pellets used.

RESULTS SECTION:

Overall, this section is OK.

Near Line 329: Figure 2 is lacking.

It is unclear whether female deer used in the experiment developed reproductive activity or it was experimentally avoided. Please clarify this point here or at Materials and methods section.

DISCUSSION SECTION:

This section is OK.

Line 374: Replace “influence” with “influenced”.

CONCLUSIONS:

Indicate that these conclusions refer to red deer, for example as follows: Line 438: “experimentally for a non-hibernating large mammal (red deer) that the seasonal changes”.

REFERENCES SECTION:

In general terms, this section is well organised and adjusted to the style of the journal for references. However, I recommend reviewing it for removing typos. Moreover, Animals uses long dash to separate page numbers of the articles: “12-23”, rather than “12-23”.

FIGURES:

Some improvements are needed:

Figure 2 is lacking.

Size of Figure 3 can be reduced, for example to 2/3 of his actual size.

Title of Figure 3 must begin below the figure.

In my opinion, Figure S1 can be placed in the manuscript, rather than as a supplementary material.

Author Response

GENERAL COMMENT:

I consider this work is within the scope of “Animals”. It contains information useful in a field in which available information is scarce. Overall, it is well written and organised. I indicate below only minor points to be improved in the manuscript.

TITLE:

It is all right.

SIMPLE SUMMARY:

It is OK.

ABSTRACT:

Remove “(1)”, “(2)”, “(3)” and “(4)” in lines 22, 26, 31 and 35.

Done, we just followed the recommendations from instructions for authors when introducing these numbers.

The Abstract must be understood independently of the other parts of the article, because it is also included in the indexing databases. Therefore, it is necessary to write with the full words the first time the following symbols and acronyms are written:  Tb, PUFA, LA, ALA, fH

Done, thanks for this suggestion.

KEYWORDS:

I recommend adding the following keyword: “cervids”.

Done, thanks for this suggestion.

INTRODUCTION:

Overall, this section is OK.

Line 52: The first time “ATP” appears in the text, must be accompanied by the full words, with ATP between brackets: “adenosine triphosphate (ATP)”.

Done

Line 83: The first time “PL” appears in the text, must be accompanied by the full words, with PL between brackets.

Done

MATERIALS AND METHODS:

Overall, this section is OK.

Line 114: Add the information of the proximal composition and nutritive value of the red deer pellets used.

Requested information is added with new Table 1 in 2.1.

RESULTS SECTION:

Overall, this section is OK.

Near Line 329: Figure 2 is lacking.

Sorry, the Figure must have been lost during the submission process. The figure was included in the file uploaded. We will now upload the figures not only in the text but also as additional files and ask the publisher to make sure that all figures are correctly included.

It is unclear whether female deer used in the experiment developed reproductive activity or it was experimentally avoided. Please clarify this point here or at Materials and methods section.

We added a sentence with information about reproduction of the experimental animals during the study (new lines 111-112).

DISCUSSION SECTION:

This section is OK.

Line 374: Replace “influence” with “influenced”.

Done

CONCLUSIONS:

Indicate that these conclusions refer to red deer, for example as follows: Line 438: “experimentally for a non-hibernating large mammal (red deer) that the seasonal changes”.

Done

REFERENCES SECTION:

In general terms, this section is well organised and adjusted to the style of the journal for references. However, I recommend reviewing it for removing typos. Moreover, Animals uses long dash to separate page numbers of the articles: “12-23”, rather than “12-23”.

Corrected

FIGURES:

Some improvements are needed:

Figure 2 is lacking.

Sorry, the Figure must have been lost during the submission process. The figure was included in the file uploaded. We will now upload the figures not only in the text but also as additional files and ask the publisher to make sure that all figures are correctly included.

Size of Figure 3 can be reduced, for example to 2/3 of his actual size.

Indeed, Figure 3 needs to be shrunk for publication. We ask the publisher to shrink or expand during final editing all Figures of the paper in a way to achieve identical size of the fonts. We will double check whether this occurred satisfactorily when receiving page proofs.

Title of Figure 3 must begin below the figure.

We agree with this objection. However, we tried to adjust the text properly in the WORD template provided by “animals”, but the position of Figures apparently changes when uploading the file. We ask the publisher to correctly place Figure titles and legends in the text during final editing. We will double check correct alignment when receiving page proofs.

In my opinion, Figure S1 can be placed in the manuscript, rather than as a supplementary material.

Done
